# Mechanisms of Arrhythmias in the Brugada Syndrome

**DOI:** 10.3390/ijms21197051

**Published:** 2020-09-25

**Authors:** Michiel Blok, Bastiaan J. Boukens

**Affiliations:** 1Department of Medical Biology, Amsterdam University Medical Center, University of Amsterdam, 1105 AZ Amsterdam, The Netherlands; 2Department of Experimental Cardiology, Amsterdam University Medical Center, University of Amsterdam, 1105 AZ Amsterdam, The Netherlands

**Keywords:** RVOT, arrhythmia, Brugada syndrome, fibrosis

## Abstract

Arrhythmias in Brugada syndrome patients originate in the right ventricular outflow tract (RVOT). Over the past few decades, the characterization of the unique anatomy and electrophysiology of the RVOT has revealed the arrhythmogenic nature of this region. However, the mechanisms that drive arrhythmias in Brugada syndrome patients remain debated as well as the exact site of their occurrence in the RVOT. Identifying the site of origin and mechanism of Brugada syndrome would greatly benefit the development of mechanism-driven treatment strategies.

## 1. Introduction

The right ventricular outflow tract (RVOT) is the preferential site of origin of arrhythmias in the setting of Brugada syndrome, idiopathic ventricular tachycardia, and to a lesser extent arrhythmogenic right ventricular cardiomyopathy (ARVC) [1,2,3,4,5,6]. Although extensive research has provided compelling evidence in support of this observation, several fundamental questions remain unanswered: what intrinsic aspects of the RVOT make it a preferential site of arrhythmogenesis, and what pathophysiological mechanisms contribute to their occurrence?

Brugada syndrome is an inherited arrhythmic disorder characterized by an increased risk of ventricular tachycardia/ fibrillation and sudden cardiac death (SCD) [7]. The disorder is named after brothers Pedro and Josep Brugada who in their study of 1992 identified a subgroup of patients with recurrent episodes of aborted SCD but did not present overt structural heart disease [8]. These patients presented the right bundle branch block, short ventricular extrasystole coupled to polymorphic ventricular tachycardia, and a type 1 electrocardiogram (ECG) pattern consisting of a normal QT-interval but ST-segment elevation in the right precordial ECG (corresponding to leads V1-V3) [9]. Because of the variable nature of Brugada syndrome ECG and the fact it can even disappear temporarily, its manifestations are commonly provoked and unmasked with the sodium channel blockers ajmaline and flecainide [10,11,12].

Brugada syndrome is associated with a myriad of pathogenic mutations [7]. Genes affected by these mutations mainly encode for subunits of cardiac sodium, calcium, and potassium channels, but also regulators of these channels [13,14,15]. Loss-of-function mutations in the gene *SCN5A*, encoding the cardiac sodium channel Na_V_1.5, are found in about 11–28% of Brugada syndrome patients but most frequent (about 62%) in the younger patients diagnosed (<17 years old) [13,16]. Currently, >300 distinct mutations have been uncovered in *SCN5A* of which most are localized in the transmembrane-spanning regions [13,17]. Even if genetically inherited, Brugada syndrome is usually diagnosed during adulthood when the ECG and arrhythmic manifestations become apparent [16,18,19]. Despite the current evidence of its extensive genetic origin, the pathophysiological mechanism that causes life-threatening arrhythmias and SCD in Brugada syndrome remains a matter of debate [7,20].

In this review, we will discuss the current understanding of the mechanism and source of arrhythmias in the respective settings of Brugada syndrome. Firstly, this review will describe the unique aspects of the RVOT with respect to its anatomical structure, developmental background, and molecular characteristics. Second, we will summarize the scientific progress and discuss important findings relevant to understanding the mechanism underlying arrhythmias in Brugada syndrome. 

## 2. The Right Ventricular Outflow Tract

### 2.1. Anatomy of the RVOT

The RVOT, also referred to as the free-standing muscular infundibular sleeve, is a thin smooth-walled tubular structure positioned between the pulmonary trunk and the right ventricular cavity, orientated cranial to the tricuspid valve (Figure 1). The RVOT can be divided into the anterior, left, right (free), and posterior (septal) walls; the latter is directly anterior to the LVOT and adjacent to the sinus of Valsalva [21]. Before birth, the process of OFT rotation results in the complex anatomical relationship between the RVOT and the LVOT. The RVOT wraps around the LVOT and their distal parts are positioned laterally relative to each other [21,22].

The RVOT wall is a multi-layered structure featuring three-dimensional networks of myofibers in a matrix of fibrous tissue. Here, the differences in the spatial orientation of these myofibers determine the complexity of the contractile motion of the heart. Compared to the right ventricle (RV) and LVOT, the spatial orientation of the RVOT myofiber bundles is different [23,24]. While its subepicardial layer is composed of circumferentially orientated myofibers, its middle and subendocardial layers are composed of myofibers running in a longitudinal direction, i.e., orientated to the long axis of the RVOT (Figure 1). This physiological variation in myofiber direction could, in theory, render the RVOT more vulnerable to myofiber disarray, particularly in the context of structural abnormalities. [25,26].

Between the left ventricle (LV) and RV, large differences exist in structural makeup including the distribution of fibrous tissue and fat. In the healthy human heart, the RV typically contains a greater collagen content compared to the LV, irrespective of age or sex [27,28]. Although in a recent report healthy tissue sections from the human RVOT were mentioned to be evaluated, unfortunately, no quantification of collagen content nor fat content in these specific sections were reported [28].

Similarly, fat is present in a healthy heart and typically most abundant in the RV wall, particularly at the level of the anterolateral and apical regions [29,30]. Besides presence at the epicardial surface, fat infiltrates the underlying myocardium (i.e., fatty infiltration) and localizes around vessels and nerves or separate strands of myocardial fibers [29]. It was found by Burke and colleagues that in human hearts with ARVC, intramyocardial fatty infiltration increased most extensively in the RVOT as well as the lateral apex [31]. Altogether, the question arises of what characteristics of RVOT render it a more vulnerable substrate to such structural abnormalities.

Using histology and two-photon microscopy, the RV wall of the mouse heart was previously found to be occupied by a higher number of non-vascular clefts, also of a bigger size, compared with the mouse LV wall [32]. This difference in tissue architecture was not accompanied by a significant difference in fibrosis levels [32]. Although the number and size of intramural clefts may affect the normal pathway of conduction in the RV and thereby cause a delay in transmural activation or conduction block, several key questions remain unanswered. Does the RV of the human heart similarly contain a higher number of intramural clefts than the LV? If the number of intramural clefts would indeed be directly proportional to the extent of conduction slowing and thus cause an arrhythmogenic substrate, does the RVOT contain more of such clefts than other anatomical structures of the heart [33]?

### 2.2. Development of the RVOT

The RVOT, like the LVOT, stems from the embryonic structure referred to as the cardiac OFT. The morphogenesis of the cardiac OFT can be summarized into the following main steps: specification and proliferation of progenitors of the second heart field (SHF), differentiation of SHF progenitors into OFT tissue, migration of cardiac neural crest cells, OFT septation, and OFT rotation.

The cardiac OFT is derived from a subpopulation of cardiac progenitor cells that comprise the SHF and is the result of their continuous proliferation and addition to the primitive, linear heart tube (Figure 2) [35]. Consequently, rightward looping of the elongating heart tube contributes to it acquiring a more complex structure reminiscent to the definitive heart [36] and is followed by the formation of the early chambers as the myocardium of the heart tube bulges out (Figure 2) [37]. Meanwhile, as the newly formed chambers differentiate and acquire a secondary, or “working”, myocardial phenotype, the OFT myocardium retains its primary myocardial phenotype. While the properties of the working myocardium are associated with high expression of genes involved in respiratory-chain function, myofibril assembly, fast impulse conduction, and the contractile apparatus, primary myocardium marks nodal cells of the pacemaker and conduction system [38]. During fetal development, the RVOT differentiates into a more working myocardium phenotype, indicated by loss of expression of Tbx2 [39]. Despite this, Cx43 expression remains extinct in part of the RVOT after birth. 

Migration and population of cardiac neural crest cells from the pharyngeal arches in the cardiac OFT contribute to its proper elongation, septation, rotation, and consequently proper arrangement of the LVOT and RVOT with respect to their respective ventricles [40,41,42].

### 2.3. Molecular and Electrophysiological Makeup of the RVOT

Electrophysiological differences can be detected from apex-to-base as well as across the ventricular wall [34,37,39,43,44,45,46,47,48,49]. This electrophysiological pattern is the result of a complex network of inductive signals and transcription factors of which the expression is spatially and temporally regulated over the extent of early embryogenesis to postnatal maturation of the heart [50]. The gene regulatory network underlying the electrophysiological patterning of the RVOT remains largely unknown [51,52]. Just before birth, most of the RVOT myocardium acquires a working phenotype except for the subregion that is located just below the pulmonary valves (Figure 1) [37,39]. In the adult mouse heart, this myocardial subregion is marked by the absence of Cx43 expression, while the rest of the adult RVOT expresses Cx43 albeit at a lower level compared to the RV [53,54,55]. Despite its confined expression in the OFT and not the ventricular wall, and function as a transcriptional suppressor of the genes encoding the fast-conducting connexins Cx40 and Cx43 [50], Tbx2 was previously found not to regulate the expression of Cx43 in the RVOT [39]. Up to date, it is not completely understood what regulatory mechanism underlies the presence of this RVOT subregion [39]. Furthermore, *SCN5A* expression is significantly lower in the adult mouse RVOT compared to the LV and RV. Although one may assume these electrophysiological characteristics would lead to conduction slowing, conduction velocity in the adult pig and mouse RVOT is similar to that in the RV [39,55,56].

While the electrophysiological heterogeneity throughout the RVOT itself may not necessarily explain why it is a preferential site of arrhythmogenesis, it may potentially render the RVOT a more susceptible region when electrophysiological changes would occur. For example, in *SCN5A*^1798insd/+^ adult mice, conduction was slowed to a larger extent in the RVOT compared to in the RV [39,57,58].

Genetic studies and subsequent functional follow-up studies have shed significant light on the key factors determining the electrophysiological characteristics of the RVOT [59]. In adult mice, heterozygous *Hey2* did not lead to abnormal ECGs nor structural heart disease. However, conduction velocity was significantly increased in the RVOT but not the RV nor LV free walls, underscoring an important role for HEY2 in determining the electrophysiological patterning of the RVOT [60]. HEY2 is a Notch-responsive transcription factor and an important regulator of cardiac muscle development, enriched in the subepicardium but not the subendocardium during embryogenesis while absent from the OFT [61,62,63]. The resultant transmural heterogeneity in HEY2 expression was shown to positively correlate with the transcriptional expression gradient of several ion channel- and gap junction-associated proteins including Na_V_1.5 [60,63,64]. In the adult mouse heart, *Hey2* expression levels become indifferent from those in the LV and RV.

More recently, the GTPase RAS associated with Diabetes (RAD) was found to be expressed four-fold higher in the mouse RVOT compared with the LV and RV [65]. In addition, human RAD was found to be expressed in a transmural pattern, being higher in the subepicardium compared to endocardium in both ventricular compartments, suggesting that the same may apply to human RVOT myocardium [65]. In guinea pig ventricular cardiomyocytes, overexpression of a dominant-negative Rad mutant led to the upregulation of L-type calcium channel expression, resulting in a larger L-type calcium current density and prolongation of action potential duration [66]. These results may imply that RAD in the human RVOT regulates a lower L-type calcium channel expression in the subepicardium, thus resulting in a subepicardial action potential that is abbreviated compared to subendocardial action potentials. 

## 3. Mechanisms Underlying Arrhythmias in Brugada Syndrome

### 3.1. The Settlement of the Debate about the Depolarization and Repolarization Hypothesis

Since the discovery of Brugada syndrome, controversy driven by a series of experimental findings supporting either of both the repolarization and conduction hypothesis caused considerable debate on what mechanism underlies Brugada syndrome pathophysiology [67,68,69,70]. In search of the cellular basis for Brugada syndrome, Yan and Antzelevitch and colleagues introduced in 1999 the concept of transmural dispersion of repolarization as the underlying cause of ST-segment elevation in the right precordial leads of the ECG [71]. Accordingly, this was due to a fundamental difference in transient outward current (*I*_to_) which is intrinsically larger in the subepicardium in comparison to the subendocardium of the RV [72,73]. In combination with a reduction in depolarizing current, as such in the concomitant presence of an *SCN5A* loss-of-function mutation, this would result in loss of action potential dome and consequently phase-2 re-entry upon arrival of a subendocardial action potential wavefront (Figure 3) [13,17,74]. Later findings demonstrated that the electrophysiological manifestations of Brugada syndrome were most marked in the area around the RVOT [75,76,77,78,79]. 

Using activation mapping of the explanted human heart from a diagnosed Brugada syndrome patient Coronel and colleagues revealed the occurrence of significant conduction slowing in the RVOT, which was associated with a high degree of fatty infiltration and fibrosis in this region [80]. However, the characteristic inverted T wave was not present in all of the leads V1–V3 [81]. Furthermore, the presence of structural abnormality in the form of fibrofatty replacement did then not fit the Brugada syndrome description but rather favored the diagnosis of ARVC, albeit a genetic background associated with ARVC could not be identified [81]. Using electroanatomic mapping, Nademanee and colleagues showed that within the anterior wall of the RVOT epicardium, radiofrequency ablation of abnormal regions displaying fractionated electrograms and conduction delay resulted in normalization of the Brugada syndrome ECG pattern in 8 out of 9 patients [82]. The late potentials observed in these electrograms were here assumed to arise as a result of delayed depolarization due to slow conduction, a notion that was supported by Sacher and colleagues several years later [83]. Szél and Antzelevitch challenged this (conduction) hypothesis by employing a pharmacological canine RV wedge model which was similar to the model used previously by Yan and Antzelevitch [71]. With this, they claimed to have successfully replicated the same electrogram fractionations as demonstrated in clinical cases [84]. Furthermore, they claimed that the occurrence of late potentials was rather due to an abbreviation in action potential duration in the subepicardium (Figure 3).

In 2017, a study by Leong and colleagues including 11 out of 13 patients tested positive for a Brugada syndrome ECG pattern after ajmaline challenge revealed that J-ST elevation positively correlated with an increasing conduction delay in the RVOT epicardium, but not in the RV [85]. Furthermore, heart rate-corrected activation recovery interval (ARI—index of myocardial action potential duration [86]) derived from the unipolar electrogram, was most prolonged in the RVOT compared to RV and LV following ajmaline challenge. However, in contrast to the earlier claimed, ARI prolongation in these cases did not correlate with the degree of J-ST elevation thus contradicting the repolarization hypothesis [87]. In response to this, Antzelevitch and Patocskai expressed their alternative interpretation of these findings in support of the repolarization hypothesis, claiming that the negligible relative delay in conduction was not sufficient to account for the conduction hypothesis [85,88]. Rather, according to abnormal conduction, relative conduction should be roughly equivalent to the duration of ST-segment elevation.

Following their editorial, Patocskai and colleagues used an experimental model, similar to that employed by Szél and Antzelevitch (see [84]), to show that the effect of ajmaline on conduction slowing was dependent on the magnitude of the action potential notch. The action potential notch was claimed to be the result of *I*_to_ activation [73,89]. Furthermore, epicardial radiofrequency ablation of RVOT regions that displayed abnormal electrograms successfully normalized the Brugada syndrome ECG pattern and abolished arrhythmic activity [89]. Furthermore, Patocskai and colleagues showed that distinct late potentials developed as a result of concealed phase 2-reentry and a delay of the second action potential upstroke which were strictly dependent on the magnitude of the action potential notch [89].

Thus far, no report fully excluded the possibility of two seemingly opposing mechanisms occurring in the same setting. A critical point of debate however remains the fact that the repolarization hypothesis derives its support, not from the true setting occurring in patients, but mainly experimental models including canine ventricular wedge preparations which were previously found to fail in reproducing the same extent of arrhythmogenesis as in intact canine hearts [69,90,91]. One of the few clinical studies in support of the repolarization hypothesis, Morita and colleagues showed that in Brugada syndrome patients, leads corresponding to the RVOT consistently showed a longer QT interval of which dispersion was increased by blocking sodium current (*I*_Na_) with pilsicainide [87,92]. Unfortunately, QT interval dispersion was not demonstrated in healthy subjects with the reason that such a control group could not be included [92].

To date, the current view based on accumulating evidence from clinical studies using electric mapping and body surface mapping strategies indicates that in Brugada syndrome patients with a type 1 ECG, (I) conduction in the RV/RVOT is slowed concomitant with significant activation delay, (II) there is an increase in the number of late potentials in the area of the RV/RVOT, and (III) changes in repolarization duration are not different compared to in healthy controls [93,94,95,96,97,98]. However, we think that in Brugada syndrome, genetic predisposition in the form of ion channel mutations involved in ventricular conduction does not solely determine arrhythmic risk but would rather modulate arrhythmic risk in the presence of 1 or more additional factors.

### 3.2. Structural Abnormalities Appear Crucial for Explaining Arrhythmias in Brugada Syndrome

In 1989, Martini and colleagues presented on six patients with ventricular fibrillation without apparent structural heart disease but J point elevation in the ECG [99]. Interestingly, these patients described by Martini fit the diagnostic criteria for Brugada syndrome and showed structural abnormalities in the myocardium after careful reevaluation, leading Martini and colleagues to suggest that RV structural abnormalities underlie Brugada syndrome [100]. Since the discovery by Martini and colleagues, multiple lines of evidence have emerged in support of the presence of structural abnormalities in the hearts of Brugada syndrome patients which altogether led to the emergence of a new hypothesis for the Brugada syndrome pathophysiological mechanism, unifying the presence of structural abnormalities with reduced cardiac excitability [80,101,102,103,104,105,106,107,108].

Considering the increased presence of structural abnormalities in the RVOT wall of the Brugada syndrome heart [80,101,102,103,104,105,106,107,108], Hoogendijk and colleagues showed in 2010 how abnormal conduction in the structurally abnormal RV/RVOT could cause Brugada syndrome (Figure 3) [102]. According to their hypothesis, ST-segment elevation is the result of a current-to-load mismatch leading to excitation failure of the RV/RVOT subepicardium located distally to sites of an isthmus or sudden myocardial expansion and conduction block, formed by structural barriers in the form of fibrosis and fatty infiltration (Figure 3). A decrease in depolarizing current available for propagation (i.e., conduction reserve [109] or safety factor for cardiac conduction [110]) would potentiate a higher number of sites of conduction block as indicated by further elevation of the ST-segment upon increasing ajmaline dose [103]. Alternatively, a decrease in long-type calcium current (*I*_CaL_) or an increase in *I*_to_ would lead to a similar outcome [110,111,112,113,114]. The same area with structurally abnormal myocardium would disrupt the normal pathway of conduction, thus setting the stage for discontinuous impulse propagation causing activation delay because of the increased pathlength and electrogram fractionation (Figure 3). Via this abnormal conduction pathway, the propagating activation front would circle these sites of conduction block and cause re-entry, provided that activation distal to the site of conduction block is sufficiently delayed [115]. In turn, this re-entrant circuit would facilitate the development of self-perpetuating rapid and abnormal activation causing ventricular tachycardia, deteriorating into ventricular fibrillation, and asystole shortly after [116]. As described earlier, catheter ablation of such abnormal areas displaying fractionated electrograms in the anterior RVOT subepicardium should be highly efficient in eliminating the Brugada syndrome ECG pattern, provided that the intervention is executed adequately [82,117,118].

In 2015, Ten Sande and colleagues elaborate on this hypothesis, providing support using patient-derived findings that fractionated electrograms and ST-segment elevation in the RVOT is likely to arise in the same structurally abnormal subepicardium of the RV/RVOT region [102,111,119]. Electrogram fractionation was present in both Brugada syndrome patients and healthy controls but was less severe and not related to ST-segment elevation in the control group, suggesting that both phenomena arise by different mechanisms. A major drawback of their study was the absence of any documented evidence supporting the presence of structural abnormalities within the RVOT myocardium of their patient group [119]. The main bottleneck, as Ten Sande and colleagues pointed out in their communication [119], is the absence of conventional imaging techniques and diagnostic modalities capable of detecting the subtle structural abnormalities expected to be present in the RVOT architecture of live Brugada syndrome patients [102].

That structural abnormalities in the RVOT myocardium indeed can lead to excitation failure was demonstrated by Vigmond and colleagues by using a wedge preparation of the right ventricular wall from the heart of a 72-year-old female case without a history of cardiovascular disease [120]. The wedge preparation showed severe structural abnormalities characterized by fatty infiltration throughout the transmural wall, thereby dividing the RVOT myocardium into separate bundles. As a result, after activation of the proximal RVOT, local epicardial electrograms at the distal RVOT epicardium displayed fractionation and ST-segment elevation. Consistent with this, simulation of structural discontinuity at the margin of the distal RVOT area led to excitation failure of the same area. Despite the presence of fatty infiltration, the absence of fibrosis suggested that the donor likely did not suffer from ARVC and this was likely just a case of an aged heart [120].

## 4. Towards a Molecular Understanding of Brugada Syndrome

### 4.1. Known Genes Involved in the Brugada Syndrome Mechanism 

Numerous mutations in, or genetics variants near, genes encoding proteins carrying ionic current have been discovered in Brugada syndrome patients [13,14,15]. Each of these genes can be seen as individual puzzle pieces which if put together reveal the puzzle that is the complex pathophysiological mechanism underlying Brugada syndrome. However, some genetic mutations or variants may occur commonly or rarely and are likely to vary in effect size. Only for a few of these genetic mutations and variants, the mechanism has been delineated. 

Recently, whole-exome sequencing analysis identified *RRAD* as a novel gene associated with Brugada syndrome susceptibility [65]. Compared to control cells from an unaffected brother, induced pluripotent stem cell (iPSC)-derived cardiomyocytes derived from the index patient showed a severe decrease in *I*_Na_, while *I*_CaL_ was moderately decreased (Figure 4) [65]. Furthermore, about 70% of these cells presented with cytoskeletal defects reflected by an abnormal round shape and a reduced count of focal adhesion points (Figure 4) [65]. 

Through a genome-wide association study of 312 patients, Bezzina and colleagues previously identified three common genetic variants at the *SCN5A-SCN10A* and *HEY2* loci [63]. The identified variant in the *HEY2* locus was associated with an increased expression of HEY2 in the heart (Figure 4) [63]. Heterozygous loss of *Hey2* in mice resulted in increased conduction velocity in the RVOT compared to wildtype mice [63]. The following study by Veerman and colleagues showed that heterozygous loss of *Hey2* resulted in the loss of heterogeneity of *I*_to_ and *I*_Na_ densities between mouse RV subepicardial and subendocardial myocytes, although *SCN5A* expression was not significantly different compared to wildtype cells [60].

For the occurrence of Brugada syndrome, two components are required: (1) reduced conduction reserve and (2) structural abnormalities caused by interstitial fibrosis. Reduced conduction reserve can be due to a reduction in *I*_Na_, and *I*_CaL_ or an increase in *I*_to_ [110]. Mutations in, or genetic variants near, genes encoding proteins carrying these currents have been found in Brugada syndrome patients [13,14,15]. Conduction reserve decreases by the administration of Class I antiarrhythmics (e.g., Ajmaline or Flecainide [12]) or by reduction of sympathetic activity [121]. In the presence of structural abnormalities, mainly caused by fibrosis, reduced conduction reserve can cause conduction delay or block [103]. The origin of interstitial fibrosis in Brugada syndrome patients is unclear. The sodium channel Na_V_1.5 may, next to its electrogenic role, be involved in the formation of fibrosis [122,123]. It is also thought that RAD affects the attachment of cardiomyocytes to the extracellular matrix resulting in gaps between myocytes and space for fibrosis to form [65]. Genes encoding proteins important for securing desmosome integrity (e.g., PKP2) predispose to structural abnormalities and reduce *I*_Na_ in ARVC patients [124]. These proteins could play a relevant role in Brugada syndrome as well [125]. Interstitial fibrosis increases with which explains why Brugada syndrome is more prevalent in older people. Moreover, we think that the physical impact on the chest, or even the shape of the thorax, may cause interstitial fibrosis.

### 4.2. The Role of The Intercalated Disk in Reducing Conduction Reserve and Formation of Fibrosis 

Despite classically defined as distinct clinical entities, Brugada syndrome and ARVC exhibit significant phenotypic overlap (for expert review, see reference [126]). In 1996, Corrado and colleagues reported an Italian family presenting ECG and clinical manifestations characteristic of Brugada syndrome [8,99], while post-mortem analysis revealed adipose replacement of the RV free wall indicative of RV cardiomyopathy [108]. Brugada syndrome and ARVC both are associated with mutations encoding proteins of the intercalated disk. The cardiac intercalated disk hosts ion channel complexes including those consisting of Na_V_1.5 channels [127]. While Brugada syndrome is associated most predominantly with loss-of-function mutations in the *SCN5A* gene (Figure 4), ARVC is a genetic myocardial disorder and most commonly associated with mutations in the *PKP2* gene encoding the desmosomal protein plakophilin-2 (PKP2], causing loss of integrity of the desmosome and disruption of structural coupling between adjacent cardiomyocytes [128,129,130]. As a result, *PKP2* mutations render cardiomyocytes more vulnerable to mechanical forces. The molecular mechanism underlying fibrotic formation and adipogenesis in ARVC likely involves PKP2 loss-of-function, given previous findings that loss of PKP2 knockdown leads to activation of signaling transduction pathways associated with transcriptional induction of genes that promote both forms of structural change [131,132].

Several *PKP2* mutations were previously identified in five unrelated individuals diagnosed with Brugada syndrome [133]. When transfected in HL-1 cells, these PKP2 mutants led to a decrease in *I*_Na_ density, likely caused by a defect of microtubule anchoring and Na_V_1.5 trafficking to the intercalated disk (Figure 4) [133]. Such cytoskeletal abnormalities may precede or even give rise to larger-scale structural abnormalities on the tissue level. Besides PKP2, the presence of Cx43 at the intercalated disk was previously shown to be a prerequisite for microtubule-dependent trafficking of Na_V_1.5 to the intercalated disk, as well as maintaining Na_V_1.5 complexes [125,134,135,136,137,138,139]. On its part, the normal distribution of Cx43 at the intercalated disk relies on the presence of functional PKP2 [139,140]. Cx43 protein levels, quantified with immunofluorescence staining, were previously shown to be reduced at the intercalated disk in RVOT sections from Brugada syndrome patients compared to control hearts [104]. Furthermore, these RVOT sections showed increased epicardial surface fibrosis with some infiltrating the underlying myocardium causing interstitial myocardial fibrosis which was accompanied by focal replacement fibrosis [104]. In respect of whole hearts, the greatest collagen content was found in the RVOT epicardium [104]. Reducing Cx43 expression by about halve of total levels was previously shown to cause a reciprocal, significant increase in collagen deposition in the ventricles of aged mice but not the young [141]. Similarly, old but not young mice heterozygous for *SCN5A* showed an increased abundance of fibrosis in both cardiac ventricles with a reciprocal disturbance of the regular Cx43 distribution in and around areas containing fibrosis [142]. More recently, the *SCN5A*^1798insD/+^ mutation was demonstrated to cause structural abnormalities in the mouse embryo, thereby impairing ventricular activation before the onset of *I*_Na_ [122]. In Brugada syndrome patients, RV ejection fraction was recently found significantly lower compared to control subjects [98]. Furthermore, the same study showed that RV/RVOT areas with abnormal electrograms significantly correlated with the occurrence of mechanical dysfunction reflected by abnormal motion and deformation leading to wall stress [98]. Over time, such wall stress may facilitate the formation of structural abnormalities [98,102]. Collectively, these findings support an important contribution of cardiac aging to the formation of structural abnormalities in myocardial areas when the electromechanical coupling between cardiomyocytes is impaired, a notion that agrees with the conduction hypothesis postulating that structural abnormalities in Brugada syndrome are acquired throughout life explaining why the Brugada syndrome ECG is rare in children and more often manifests later in life [16,18,19,102].

Altogether, accumulating evidence has provided support to the notion that disruption of desmosome integrity (I) affects *I*_Na_ density, evidenced by a decreased amplitude of the *I*_Na_ in cardiomyocytes, (II) facilitates *I*_Na_-dependent lethal arrhythmias in vivo, and (III) is likely to provide at least part of the molecular substrate of Brugada syndrome [125,133,134,143]. Furthermore, these findings should encourage the inclusion of *PKP2* as part of routine genetic testing for Brugada syndrome, in particular, to promote a better understanding of the key players participating in the molecular mechanism of this disorder. However, given the previous findings by Coronel and colleagues, the absence of an ARVC-associated genetic background whilst the presence of structural abnormalities suggests that desmosomal defects associated with ARVC are not necessarily a prerequisite for the occurrence of structural abnormalities in Brugada syndrome [80,81].

### 4.3. Insights From Human Induced Pluripotent Stem Cell Models

The human iPSC technology has consistently been employed over the years to allow investigation of the molecular and cellular mechanism of Brugada syndrome in the setting of the native patient-specific cell environment. Despite this major advantage, human iPSC-derived cardiomyocyte models fail to capture the complex changes in tissue architecture that occur in Brugada syndrome. Nevertheless, human iPSC-derived cardiomyocytes are a useful tool to study the functional predisposition to Brugada syndrome. Collectively, these studies led to profound divergent results which could be explained by the variety of genetic mutations studied. In part, the effect of mutations may vary, even if affecting the same gene [144,145]. Furthermore, protocols for human iPSC-derived cardiomyocyte differentiation do not yield a pure cell population of a single type, but rather a variety of cardiomyocytes with divergent phenotypes. In addition, human iPSC-derived cardiomyocytes are characterized by their immature, fetal-like phenotype which, compared to adult cardiomyocytes, consists of different structural and functional properties [146].

While some studies reported no clear electrophysiological abnormalities in Brugada syndrome patient-derived iPSC-derived cardiomyocyte lines compared to controls [147,148], other reports contrarily showed evidence of significant alterations in action potential duration [65,144], decreased *I*_Na_ density [65,144,145,149,150,151], decreased action potential upstroke velocity [65,144,145,150,151], or irregular calcium handling [65,145]. Ma and colleagues found that pacing at a frequency of 0.1 Hz led to a small subgroup (25%) of Brugada syndrome patient-derived iPSC-derived cardiomyocytes presenting with action potentials which were by the authors claimed to resemble the loss of action potential dome configuration as postulated by the repolarization hypothesis, albeit the pacing frequency exceeding human physiological range [71,144]. Only a few studies focus on the potential presence of morphological changes, which were observed by Belbachir and colleagues in the form of cytoskeletal defects [65].

## 5. Summary

The scientific debate on the mechanism of Brugada syndrome has led to significant advancement in our understanding of its manifestations. The identification of structural abnormalities in the RVOT myocardium of Brugada syndrome patients has led to the current view that abnormal conduction as the pathophysiological mechanism. A multitude of factors would likely contribute to the presence of these structural abnormalities. Improvement of diagnostic imaging modalities and techniques could allow such subtle abnormalities to be detected in the hearts of Brugada syndrome patients both to help further progress the field and also to improve the management of Brugada syndrome patients. The current understanding of both ARVC and Brugada syndrome phenotypes hint toward a partial overlap in pathophysiological mechanisms causing structural abnormalities as seen in both disorders. Further investigation into this concept would improve the understanding of the underlying pathophysiology of the Brugada syndrome.

In conclusion, the RVOT is a unique structure in terms of anatomy and electrophysiology, which could explain why its preferential site of origin in the setting of Brugada syndrome. However, although various hypotheses have presented themselves over the last decades, the true answer remains concealed. 

## Figures and Tables

**Figure 1 ijms-21-07051-f001:**
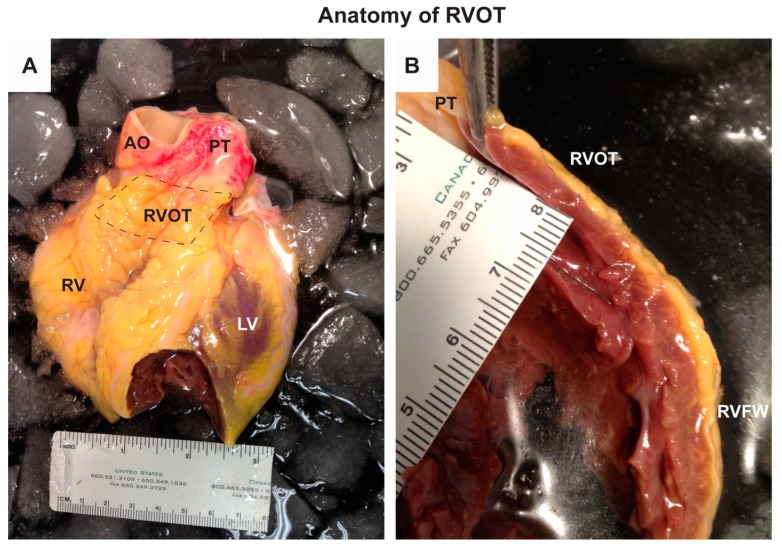
Anatomy of the right ventricular outflow tract (RVOT). The photographs are from a human heart of a male in his mid-fifties and taken by BJB. Theheart was provided by Mid-America Transplant Services (St. Louis, MO, USA), as described previously [34], and its use was approved by the Washington University School of Medicine Ethics Committee [Institutional Review Board (IRB)]. Panel A shows the anterior side of the heart with RVOT indicated by the dashed box. Panel B is a photograph of a cross-section of the right ventricle showing the RVOT at the top. Note the absence of the trabecles at the lumen of the RVOT. RV, right ventricle; RVFW, right ventricular free wall; LV, left ventricle; AO, aorta; PT, pulmonary trunk.

**Figure 2 ijms-21-07051-f002:**
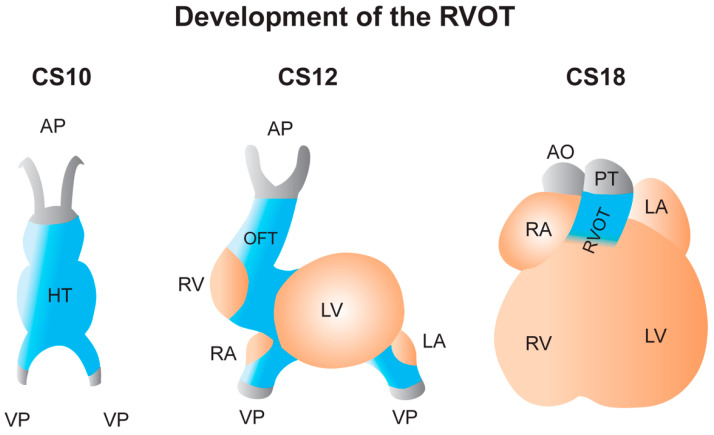
Development of the RVOT. Primary myocardium is indicated in blue and the embryonic working myocardium in orange. The embryonic outflow tract (OFT, composed of primary myocardium) gives rise to the (future) RVOT in the fetal and adult heart. During Carnegie stage (CS) 10, i.e., after 28 days of development, the arterial pole and venous poles of heart tube (HT) elongate by continuous proliferation and addition of cardiac progenitor cells. It is from CS10 that rightward looping of the linear heart tube initiates. During CS12 (i.e., day 30), as the heart tube loops, the cardiac chambers start to balloon out. At this stage, further elongation of the arterial pole has given rise to the tubular outflow tract (OFT) which connects the right ventricle (RV) to the aortic sac and aortic arches. During CS16 (i.e., day 39), the OFT has become relatively shorter with concomitant incorporation of its proximal part into the RV. Septation of the proximal OFT initiates at CS14, proceeds during CS16 by septation of the distal OFT, thereby giving rise to the pulmonary and aortic channels at CS18. At CS18 (i.e., day 44), the OFT is configured as for the postnatal heart. Both pulmonary and arterial components of the OFT are relative to each other in a spiral orientation. AP, arterial pole; VP, venous pole; LV, left ventricle; LA, left atrium; RA, right atrium; RVOT, right ventricular outflow tract; AO, aorta; PT, pulmonary trunk.

**Figure 3 ijms-21-07051-f003:**
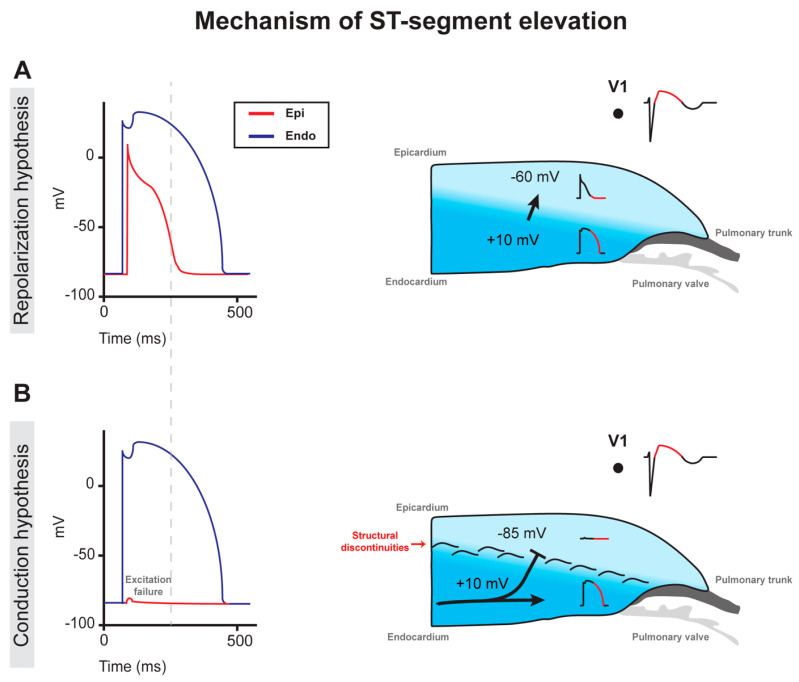
Mechanism of the ST-segment elevation in Brugada syndrome as postulated by the repolarization and conduction hypotheses. Action potentials from the subendocardial and subepicardial myocytes are indicated in blue and red respectively. (**A**) Transmural dispersion of repolarization, characterized by a loss of action potential dome in RVOT subepicardial myocytes, is caused by a reduced sodium current (*I*_Na_) in combination with strong intrinsic expression of the transient outward current (*I*_to_). (**B**) An abnormal pathway of conduction, imposed by subtle structural abnormalities throughout the RVOT myocardium, causes a current-to-load mismatch and conduction block which, in turn, leads to excitation failure of subepicardial myocytes.

**Figure 4 ijms-21-07051-f004:**
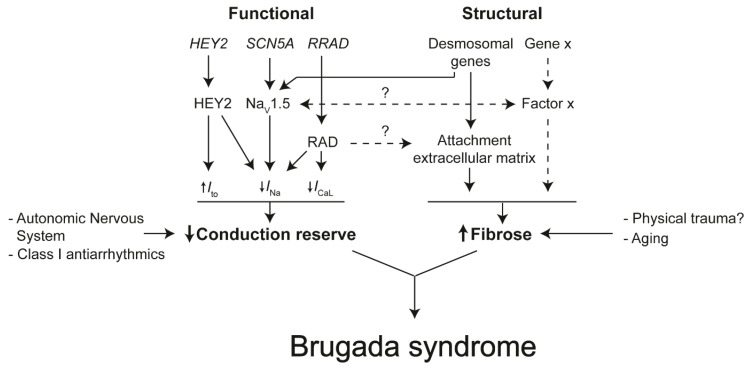
The working hypothesis underlying Brugada syndrome.

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
