# Peer review of "Mechanisms of Arrhythmias in the Brugada Syndrome"

_ijms, 2020, doi:10.3390/ijms21197051_

Round 1
Reviewer 1 Report
The authors are to be congratulated for an excellent paper.
Some points remain:
Page 6, line 219, reword.
Page 7 - It has been shown that Brugada syndrome is associated with a lower RV ejection fraction, reduced mechanical function of the RVOT anterior free wall, reduced gap junction expression, and increased collagen.
Page 7 - "hypothesis of abnormal conduction as the pathophysiological mechanism": since the very definition of Brugada syndrome is an ECG abnormality, it is probably safe to say that abnormal conduction is certainly the cause of the Brugada syndrome. The question is how does this abnormal conduction originate? Is it only the SCN5A gene? Is it also other genes? What sorts of effects do genetic mutations have on the molecular pathways and conduction imbalance? How is this abnormal signal propagated to other cells and maintained, deteriorating into VT/VF? Does ablation seem to fix it? Why?
Reviewer 2 Report
The cause of the preferential occurrence of idiopathic ventricular tachycardias and of ventricular tachycardias in Brugada syndrome from the right ventricular outflow tract (RVOT) is unknown. This review focusses on the anatomy, developmental and molecular characteristics of the RVOT in an attempt to explain this regional susceptibility to arrhythmias.
This review is generally well written and structured. I would like to share the following observations and suggestions.
The structure of part 4. (Arrhythmias in Brugada syndrome and the proposed mechanisms) follows a chronological line which feels outdated to this reviewer. Over the last couple of years much epicardial mapping data of the arrhythmic substrate of the Brugada syndrome patients have become available which levels of evidence is much greater than that of tissue models. The authors should use the available human data as starting point instead of theoretical models. This would in this reviewers opinion also balance this part more which currently focuses too much on the repolarization hypothesis that cannot adequately explain the fractionated electrograms found during mapping and ablation procedures let alone low-voltage areas.
This reviewer noticed that the part on the anatomy focusses mainly on the fiber direction in the RVOT. Although the RVOT is usually not studied as a separate areas it is known that the right ventricle contains more structural discontinuities such as fatty infiltration (Tansey et al. Histopathology. 2005;46:98-104) and fibrous tissue (Oken et al. Circulation Research 1957;5:357-61) than the left ventricle. As structural discontinuities are part of the mechanistic discussion later in the paper this would appear worth mentioning in this part of the paper.
Lastly, the authors unintendedly use teleological arguments at two points in the text. Part 3. Line 20: “Up to date, the functional reason…” There is no functional reason as there is no intelligent design. Part 4.2. Line 279: “The question remains for what functional purpose…” There is no functional purpose as there is no intelligent design, just situation and consequence.
Specific comments:
Introduction (line 22): Reference 1-4 do not all correspond with sentence. Reference 2 and 4 concern ARVC and LQTS not Brugada syndrome or idiopathic VTs. The best article demonstrating the involvement of the RVOT in the initiation of ventricular fibrillation in Brugada syndrome is, in this reviewers opinion, Morita et al. Heart Rhythm 2009;6:487-92.
Introduction (line 23): “this concept” consider the use of “this observation”.
Introduction (line 31): “persistent ST-segment elevation”. One of the characteristics of Brugada syndrome is that the ST-segment elevation can change and even temporarily disappear. Please remove persistent.
Anatomy of the RVOT (line 51): “caudally” is the RVOT not positioned cranial to the TV?
Round 2
Reviewer 1 Report
The authors have made substantial improvements to the paper and are to be congratulated again. All of my comments have been adequately responded to.
Reviewer 2 Report
The authors have adequately addressed my suggestions. This reviewer has no further suggestions beside the correction of some minor errors (Line 291: "atrial fibrillation" should read "ventricular fibrillation" or "ventricular arrhythmias")